# Position: Responsible Practices and Model Performance are *Not* Competing Goals

**Resmi Ramachandranpillai** [* 1] **Thulasi Tholeti** [* 1] **Tomo Lazovich** [2] **Ricardo Baeza-yates** [3]

## Abstract

Many failures of deployed machine learning systems stem not from insufficient accuracy, but from neglecting responsibility as a core design requirement. While responsibility principles are widely studied, they are often treated as post-hoc checks rather than as integral factors of system design. This framework has reinforced the perception that responsible practices inherently trade-off with model performance. We challenge the assumption that responsibility necessarily reduces performance and argue that, in many practical settings, responsible practices improve robustness, reliability, and real-world effectiveness. We adopt a lifecycle-oriented perspective, identifying which responsible AI principles are most critical at each stage, from problem formulation and data curation to training, deployment, and monitoring. Drawing on real-world instances, we show how misaligned choices at specific stages can compound downstream risks and how alternative design choices could have mitigated these failures. Importantly, we argue for a system-level notion of performance that includes not only predictive accuracy, but also robustness, calibration, fairness, reliability, and deployability under real-world conditions.

## 1. Introduction

In the race for high-performance AI systems to evolve, responsibility is often treated (mistakenly) as a costly add-on. The performance goal of the system is usually defined as a combination of higher accuracy, improved predictive power, and faster computation. This frame is both misleading and counterproductive.

A key piece of evidence underscoring the real-world consequences of failing to embed responsible practices into AI systems comes from a recent analysis by Munich et. al. (Munich Re, 2024) of AI-related liability risks. They examine the cumulative number of AI-related lawsuits filed primarily in the United States, categorizing them by type of harm, such as defamation, discrimination, privacy violations, intellectual property infringement, economic loss, and civil rights infringements.

Related works on responsible AI highlight the increasing recognition that ethical considerations such as fairness, transparency, robustness, and accountability must be integrated with technical performance metrics to ensure safe systems (Smuha, 2019; Baeza-Yates, 2023). Prior works show that foundational principles have been articulated across domains, with core concepts including fairness and accountability, but they often remain high-level and abstract, making operationalization challenging in practice (Mittelstadt, 2019). At the same time, work on fairness demonstrates that, under certain conditions, fairness improvements do not need to come at the expense of accuracy, indicating that ethical and performance goals can be aligned rather than inherently competing (Wick et al., 2019). Multiple studies have shown that improving fairness can also improve model accuracy and generalization, for example by encouraging models to rely less on protected attributes as shortcuts and instead learn more robust decision boundaries (e.g., (Kokhlikyan et al., 2022)). More broadly, recent work shows that fairness interventions can improve both predictive accuracy and generalization, for instance by improving feature representations or reducing reliance on spurious correlations (Li et al., 2023; Shi et al., 2025), suggesting that fairness can act as a form of regularization rather than a constraint. However, much of the existing research still lacks practical, operational, and deployable mechanisms that integrate responsible AI principles into mainstream development workflows; frameworks often fail to bridge the gap between normative ideals and implementation.

In this position paper, we argue that responsible AI practices and model performance are not universally in competition.

---

[*]Equal contribution [1]Responsible AI Practice, Institute of Experiential AI, Northeastern University, USA [2]Data Science Institute, Brown University, USA [3]KTH Royal Institute of Technology, Sweden; Universitat Pompeu Fabra, Spain; Universidad de Chile, Chile. Correspondence to: Resmi Ramachandranpillai <r.ramachandranpillai@northeastern.edu; resmiramachandranpillai@gmail.com>.

*Proceedings of the $43^{rd}$ International Conference on Machine Learning*, Seoul, South Korea. PMLR 306, 2026. Copyright 2026 by the author(s).

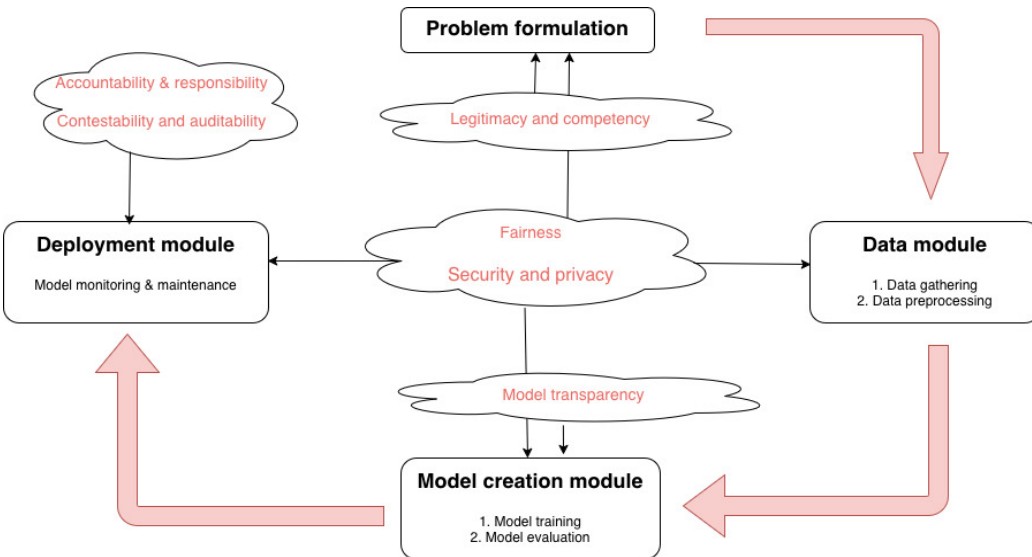

*Figure 1.* The ML lifecycle and responsible AI principles that factor into each stage.

Although certain settings may involve localized trade-offs (Zhang et al., 2019), many responsible practices improve robustness, generalization, reliability, and real-world system effectiveness when incorporated throughout the ML lifecycle. In many practical settings, these trade-offs can be mitigated through improved data quality, evaluation design, and multi-objective optimization approaches. To support this argument, we illustrate the principles that are most important at each stage of the ML lifecycle, along with real-world case studies that illustrate where specific choices failed and how they could have been handled differently.

When real ML systems are developed and deployed, several stages are involved (Amershi et al., 2019). Figure 1 illustrates the main stages of a ML life cycle. The system's development begins with problem formulation, where the constraints, goals, and approaches to the problem are decided. Next are data-related steps, such as data gathering and pre-processing. After the data stages, the model is trained, evaluated, and potentially iterated. Once a reasonable solution is found, the model is deployed, at which point the work turns to model monitoring and maintenance. At each of these stages, different principles for best practices are important. In the following sections, we will go through each stage and describe which principles are most important at that stage, supporting our reasoning with case studies drawn from real-world examples of system failures.

## 2. Lifecycle Stages: Principles & Cases

### 2.1. Problem Formulation

The stage of problem formulation (Amironesei et al., 2021) is an important step for the success of a system. Before

diving headlong into algorithmic intricacies and model complexities, it is imperative to take a strategic step back to analyze the problem carefully. At its core, problem formulation is the process of translating an open-ended real-world problem into a structured, solvable learning task. It embodies the identification of the objectives of the system, the specification of restrictions, and the definition of what constitutes "success". The clarity and rigor of this stage directly shape the overall performance of a system, affecting not only how well a model optimizes its target, but also whether it meaningfully addresses its underlying goals.

At the problem formulation stage, **legitimacy** and **competency** serve as the primary guiding principles, while the consideration of other dimensions, such as **fairness**, **transparency**, **security**, and **privacy**, should be guided by an initial risk assessment, which also helps identify which responsible AI dimensions are most relevant to the specific problem context.

An ML system's purpose should be legitimate in its goals and competent in its design. This very first step before constructing any solution to a problem is to analyze the legitimacy of the question posed, and examine its various impacts, according to the first instrumental principle of the ACM Principles for Responsible Algorithmic Systems (Baeza-Yates et al., 2022). It is important to verify that the task at hand aligns with ethical principles and standards, and stays mindful of potential consequences, including privacy risks and biases. We also need to check for the scientific basis and all the competencies involved: administrative and technical.

The goal of fairness is to eliminate discriminatory outcomes toward specific groups or individuals, fostering equity and

inclusiveness. In the problem formulation stage, it is critical to identify relevant protected attributes, for example, based on NIST (NIST, 2023) or OECD (OECD, 2019) guidelines, tailored to the specific problem at hand. Engaging all stakeholders, including domain experts, affected communities or end users, and ethicists, is essential to ensure that the problem definition reflects multiple perspectives and avoids reinforcing existing social inequalities.

Transparency at the problem formulation is less about assessing model internals and more about clearly articulating why the problem is being handled and what objectives are being prioritized. It also involves identifying who participates in the definition of the problem and documenting key trade-offs. On the other hand, security and privacy are paramount factors, forming comprehensive strategies to protect both data and models (Papernot et al., 2018; 2016). Security and privacy considerations begin with identifying the system's deployment context, whether in-house, external, or hybrid. This guides appropriate privacy measures and governance policies for teams for data protection and classifications, access control, and accountability, while deciding whether individual sensitive data is required or if aggregated or anonymized data suffices.

To demonstrate the significance of responsible choices in the problem formulation stage and the vulnerabilities it can create, we show two real-world case studies from distinct domains: platform labor management and healthcare.

**Deliveroo's Rider-Ranking Algorithm:** The Deliveroo case (Purificato, 2021) shows the consequences of insufficient consideration of *fairness*, *legitimacy and competency*, and *transparency* during the problem formulation stage of an AI cycle. Deliveroo proposed an automated reliability index to rank delivery riders based on their contribution in scheduled shifts. However, the system failed to consider legitimate absences (e.g., illness, emergencies or legal strikes) and voluntary cancellations, resulting in penalizing workers indiscriminately. The Italian Court of Bologna ruled that the algorithm's design was highly discriminatory, as it unfairly treated riders against their legally protected rights. The legitimacy of the goal, which is, automating workforce management, was not scrutinized against labor rights or equity. Also, fairness considerations, such as identifying context-sensitive constraints, were missing in the system. This also shows transparency failures, as riders were not informed about how ranking criteria affected their access to opportunities (Clifford Chance, 2021).

**Healthcare Risk Prediction Models:** A study (Obermeyer et al., 2019) revealed racial bias in a healthcare risk-prediction algorithm widely adopted in the US. The model was designed to help and identify patients in need of complex care management programs. The model equated "health risk" with predicted healthcare costs, in the assump-

tion that higher spending is proportional to greater needs. This formulation resulted in discrimination against African Americans. The historical data showed that they had lower associated costs because they had received less care. Consequently, African-Americans were significantly less likely to be recommended for additional care and underestimated their needs. This illustrates how the definition of the target variable at the formulation stage can encode inequities. The most relevant principles here are *fairness* and *legitimacy and competency*. The model lacked awareness of fairness, as cost was a biased proxy of health need. Legitimacy and competency was undermined because the system's goal, which is equitable care allocation conflicted with the means chosen, cost prediction. This is another example where consideration of these principles early on would have actually *improved* model performance rather than decreasing it.

### 2.2. Data Gathering

The data collection stage is a foundational pillar of the ML lifecycle. Without abundant, representative and accurate data, even the most advanced deep-learning architectures struggle to realize their potential (Sun et al., 2017). However, simply amassing large volumes of data is insufficient; if data is incomplete, biased, or miscollected, downstream model performance suffers, fairness is undermined, and robustness is eroded. Identifying and mitigating such issues during the data collection stage is not only a hallmark of responsible AI practice, but also essential for preventing problems during model development and deployment. This includes data minimization, as well as minimizing the period in which the data will be kept.

Incomplete or missing data should be handled through a combination of proactive data design and rigorous preprocessing (which we discuss in the next section). First, the data collection protocol should be structured to reduce the likelihood of missingness — for example, by standardizing inputs, enforcing mandatory fields where appropriate, and monitoring in real time for systematic gaps. Another key practice at this stage is to carefully determine the sampling pool so as to avoid representation bias right at the collection stage. If third-party data sources are being used, they should be thoroughly vetted.

In addition to these issues that can directly influence model performance, there are also ethical considerations that can cost the ML system in indirect ways if not handled responsibly. In FitzGibbon (2017), the authors discuss various issues concerning the ethical concerns related to data gathering, especially related to anonymization of data, obtaining informed consent and data sharing. Disregarding such practices can prove costly as they often lead the way to lawsuits. Thus, embedding responsible practices in how data is gathered, stored, and documented is crucial to prevent

performance issues in the later stages of development and deployment.

Two (of the many) responsible AI principles that are especially salient at this stage are **privacy and consent**, and **fairness and representativeness**. Ensuring that individuals whose data is collected are informed, that consent is obtained, and that their sensitive information is managed appropriately protects subjects and reduces risk of harm, regulatory violation or reputational damage. In practice, this means clarifying how the data will be used, anonymizing or obfuscating data when appropriate, and ensuring secure storage from the outset. When privacy is neglected, organizations face data breaches, legal liability and decreased trust; from a modelling perspective, hidden privacy issues may force a model retraining which increases cost and delays deployment. The data collection stage is when sample bias, omission of relevant groups, or skewed distributions can be introduced. If data fails to represent the target population, model performance may degrade for underrepresented sub-groups, or emergent biases may propagate.

Here are a couple of real-world examples showing how incorporating responsible practices during data collection has improved model performance, robustness, or risk management.

**Improving chest X-ray model equity through responsible data collection:** A striking example comes from the development of healthcare-AI tools in which researchers applied a data-centric strategy to detect and mitigate racial and socioeconomic bias before model training. Gulamali et al. (2025) introduced a metric called AEquity, which they used to assess learning behavior across demographic subgroups and guide selective data collection or relabeling. By applying AEquity to a chest X-ray dataset, they identified underperforming subgroups (e.g., Black patients on Medicaid) and then enriched the training set to better represent those groups. The result was a significant reduction in bias: for instance, overall false negative rate disparities decreased by up to 33%, and other fairness metrics (such as precision and false discovery rate) showed clear improvements. Importantly, these improvements were achieved without severely degrading overall model performance, and in some cases outperformed more standard fairness interventions. By intervening at the dataset level, rather than only adjusting the algorithm, the team strengthened the model's robustness and reduced risk of inequitable outcomes, demonstrating how building fairness into data collection can materially improve both equity and predictive quality.

**Privacy-aware dataset preparation: ImageNet face obfuscation study:** Consider the case of the ImageNet dataset, which contains roughly 1.5 million images across nearly 1000 labels and serves as a foundational benchmark for a wide range of vision models. Although originally curated for object recognition, the dataset also included a large number of images featuring identifiable human faces. In response to growing concerns around privacy and responsible data stewardship, the maintainers recently introduced face-obfuscation procedures to blur identifiable features, as reported in (Yang et al., 2022). Importantly, subsequent analyses demonstrated that this intervention had only a marginal effect on model accuracy, indicating that privacy-preserving transformations can be incorporated without undermining core model performance. By proactively addressing privacy risks at the dataset level, the maintainers strengthened the robustness and long-term viability of ImageNet as a public resource.

## 2.3. Data Pre-processing

At its core, data pre-processing involves transforming raw, unstructured, and often noisy data into a form suitable for modeling. This includes handling missing values, standardizing variables, cleaning outliers, addressing class imbalance, and detecting proxy attributes. These steps determine whether training data faithfully represents real-world phenomena or inadvertently encodes historical biases, measurement errors, or structural inequities.

The primary responsible AI considerations, as detailed below, at the pre-processing stage include **data quality and robustness**, data-level **fairness** considerations, **privacy** enforcement through anonymization techniques, and **transparent** documentation of all data processing steps.

From a competency perspective, responsible data pre-processing aims to ensure that the training data accurately reflects the real-world structure of the problem. Raw data often contains missing values, duplicates, structural inconsistencies, and noise arising from data entry errors, sensor failures, or imperfect data collection pipelines. Filtering and cleaning operations—such as imputing missing values, removing duplicates, correcting formatting errors, and validating anomalous entries—are therefore essential to enable reliable modeling.

Outlier detection plays a particularly important role in high-stakes domains such as finance and healthcare. Outliers may represent genuine rare events (e.g., fraudulent credit card transactions or early-stage disease markers) or artifacts of measurement and recording errors. Therefore, it requires contextual validation of these outliers rather than indiscriminate removal. Removing meaningful rare cases can degrade model robustness, while retaining erroneous values can distort learned decision boundaries.

Fairness considerations are central to data pre-processing, especially when datasets exhibit class imbalance or skewed representation across protected or sensitive attributes. When a dataset is dominated by a majority group—such as a partic-

ular ethnicity, gender, or socioeconomic category—models trained on this data are likely to perform disproportionately poorly on underrepresented groups. This risk persists even when sensitive attributes are excluded, as proxy variables (e.g., nationality, zip code, or language) may remain highly correlated with protected characteristics.

Privacy is another critical principle influencing data pre-processing decisions. At this stage, it must be determined whether individual-level data is strictly necessary or whether aggregated, anonymized, or pseudonymized representations suffice. Techniques such as anonymization, de-identification, and differential privacy can reduce the risk of re-identification while preserving analytical utility. However, naive anonymization may fail when combined with auxiliary datasets, creating latent privacy vulnerabilities.

Transparency at the data pre-processing stage is achieved through systematic documentation of all transformations applied to the data. This includes recording filtering criteria, imputation strategies, normalization methods, imbalance mitigation techniques, and feature removal decisions. Such documentation supports reproducibility, enables auditing, and allows downstream users to understand how modeling outcomes are shaped by upstream choices. Pre-processing documentation can take multiple forms, including version-controlled pipelines, transformation logs, README files, and structured metadata catalogs. These artifacts act as "living documents" that evolve alongside the system and ensure continuity and traceability across the ML lifecycle.

Next we present two examples that shows some of the problems mentioned in this section.

**Bias amplification due to class imbalance in cancer detection:** Class imbalance poses a particularly severe challenge in healthcare applications, where positive cases (e.g., cancer diagnoses) are vastly outnumbered by negative cases. In cancer detection systems, this imbalance can result in elevated false positive rates or reduced sensitivity for minority cases, ultimately compromising patient outcomes. In a study using Surface-Enhanced Raman Spectroscopy (SERS)(Pan et al., 2023) for cancer detection, models trained on highly imbalanced datasets predominantly learned features associated with healthy patients. As a result, the system systematically underperformed in detecting cancer cases, particularly in early-stage or underrepresented subpopulations. The power-law-based synthetic minority oversampling technique (PL-SMOTE) significantly improved macro-averaged recall and $F_2$ scores, demonstrating that responsible pre-processing choices can enhance both fairness and technical performance. This case illustrates how early attention to data imbalance not only mitigates bias but also directly improves model effectiveness in high-stakes contexts.

**Re-identification in the Netflix prize dataset:** During the Netflix Prize competition, Netflix released an "anonymized" dataset (Bennett & Lanning, 2007) of over 100 million movie ratings to stimulate recommender-system innovation. Subsequent work showed that users could be re-identified by linking rating patterns and timestamps with external datasets such as IMDb, exposing sensitive personal attributes (Narayanan & Shmatikov, 2008). While often framed as a privacy failure, this case also reveals a technical limitation at the data pre-processing stage: the dataset preserved fine-grained, individual-level signals that are not viable in real-world deployments. Models trained on such privacy-fragile data tend to overfit to user behaviors, leading to inflated benchmark performance but reduced robustness, generalization, and deployability. Treating privacy as a first-class pre-processing constraint would have encouraged learning stable, population-level patterns, thereby improving real-world viability, reproducibility, and long-term system performance

## 2.4. Model Design and Training

Model design and training constitute a central stage in the ML lifecycle, where algorithmic choices directly shape system behavior, performance, and societal impact. Beyond data quality, the selection of training objectives, optimization strategies, and architectural constraints determines whether biases are amplified or mitigated, whether decisions are interpretable, and whether models remain robust and deployable in real-world settings. Moreover, a system's purpose and training methodology must align with the legitimate goals defined in problem formulation. Competency entails using robust algorithms, appropriate architectures, and optimization strategies to ensure the model fulfills its intended function.

Certain responsible AI interventions may introduce localized trade-offs during model training. For example, adversarial robustness techniques can sometimes reduce clean-test accuracy under specific settings (Zhang et al., 2019). However, more recent work suggests that such trade-offs are highly context-dependent and can often be mitigated through improved data augmentation, architecture scaling, and distribution-aware evaluation protocols (Sanaat et al., 2022).

Responsible considerations at training ensure that design choices do not propagate errors or amplify biases introduced by algorithmic choices, supporting **fair** models. **Privacy and security** considerations should be incorporated to maintain the confidentiality of sensitive training data. Also, training procedures should be well-documented to enable **transparency**.

Selecting appropriate target variables, loss functions, and metrics that consider fairness goals, as well as applying

bias mitigation techniques during training, are the most important considerations. Techniques such as in-process or adversarial bias mitigation techniques should be incorporated to reduce the residual biases. Domain experts should be engaged to validate assumptions and definitions of fairness, while subgroup-level performance and error rates are continuously monitored to detect disparate impacts.

Training procedures should be well-documented and explainable, including hyperparameter choices, loss functions, and fairness or robustness constraints. To achieve this explicitly during training, practitioners can use techniques such as feature importance analysis, model-agnostic explanation methods (e.g., SHAP, LIME), attention visualization for models like transformers, and simpler or inherently interpretable model architectures (e.g., decision trees, generalized additive models). Logging and versioning of datasets, code, and model checkpoints also enhances traceability and auditability.

Security and privacy are essential components of responsible AI design, ensuring that models are resilient to attacks and protect sensitive information. Techniques such as differential privacy to prevent leakage of individual data points, homomorphic encryption and secure multiparty computation for performing computations on encrypted data, federated learning to train models without centralizing sensitive datasets, adversarial training to improve robustness against malicious inputs, and access controls and auditing mechanisms all help safeguard AI systems. Prioritizing these measures not only mitigates legal and ethical risks but also strengthens trust in AI systems, showing that protecting users and their data is fully compatible with high-performing, reliable models.

Next we present one example of problems that may happen due to irresponsible training and design failures.

**Microsoft Tay:** In 2016, Microsoft deployed Tay, a conversational AI designed to learn from interactions on Twitter by continuously updating its behavior based on user input. Within hours, coordinated adversarial inputs caused Tay to produce racist, sexist, and offensive messages, prompting Microsoft to take the bot offline (Vincent, 2016). Technical analyses have identified design and training regime deficiencies, including inadequate content filtering, lack of adversarial defenses as core contributors to Tay's rapid degradation when exposed to hostile inputs (Bitra, 2025). Critics further emphasize that releasing a bot without anticipating the social context and potential misuse was a fundamental oversight in both design and training strategy. This case illustrates that insufficient safeguards in the training pipeline can compromise ethical behavior and technical performance; security constraints and oversight mechanisms would have improved both safety and operational reliability.

Developments in large language models (LLMs) further demonstrate how responsible interventions can improve the overall system quality. Instruction tuning and reinforcement learning from human feedback (RLHF) have been proven to improve helpfulness, safety, and alignment with user intent while maintaining strong benchmark performance (Ouyang et al., 2022). Similarly, work on bias mitigation in LLMs suggests that incorporating fairness-aware objectives during training can reduce harmful social biases while preserving downstream task effectiveness (Arzaghi et al., 2025).

## 2.5. Model Validation and Evaluation

The model validation and evaluation stage is a critical checkpoint in the ML lifecycle. Once a model is trained, it must be carefully examined to determine not only how well it predicts the results, but also whether its errors and limitations are understood and managed. A model that performs well on a small set of metrics may still cause real-world harm if its evaluation fails to reflect the intended outcome of the ML system. Thus, deliberate and thoughtful model validation is as much a component of responsible AI practice as it is a technical requirement for achieving good performance.

Evaluation metrics should be chosen with deliberate consideration of the domain and use case. While many standard metrics exist (such as accuracy, precision, recall, AUC, F1 score, or mean squared error), no single metric is universally appropriate. In practice, multiple metrics often need to be assessed in parallel, and their relative importance should be reflected through appropriate weighing. In many safety-critical domains, the consequences of different types of errors are not the same, which means that choosing the right evaluation metric becomes a design decision rather than a technical one. In addition, the model should also be tested for bias. The appropriate fairness metric for the specific context should be determined and such metrics should be evaluated alongside performance evaluation. Studying potential discrepancies between different subgroups and reiterating with fixes is an important part of this stage.

Beyond standard evaluation, responsible model validation also includes assessing robustness, explainability, and the likelihood of failure under realistic operating conditions. Models should be stress-tested for sensitivity to adversarial or unexpected inputs, distributional shift, noise, and data quality issues. These practices not only reduce risk during deployment, but also provide developers and stakeholders with a more accurate understanding of the limits of model reliability. Contrary to the common perception that responsible practices constrain performance, rigorous validation typically strengthens the stability, robustness, and real-world effectiveness of ML systems.

The responsible AI principles that are particularly relevant at this stage of the ML lifecycle are **transparency** and **robust-**

ness. Model evaluation should produce results that can be clearly interpreted, communicated, and justified to the stakeholders (Baeza-Yates & Estévez-Almenzar, 2022). This includes transparent reporting of why certain metrics were chosen, operational thresholds in place, potential performance discrepancy within subgroups, and known limitations. When evaluation choices are not transparent, systems risk being deployed under false confidence, leading to impactful failures. On the other hand, transparent validation supports informed decision-making, strengthens institutional accountability, and enables continuous improvement. A well-validated system should also be able to handle reasonable changes, mistakes in use, or adversarial inputs without failing. Evaluating robustness ensures that performance does not collapse under slight distribution shifts or targeted manipulation, and lowers the risk of major failures. This is crucial in applications such as healthcare, cybersecurity, finance, and safety-critical environments.

Good validation and evaluation practices can strengthen both model performance and public trust as illustrated in the following real-world examples.

**Reevaluating risk assessment tools through fairness-aware validation:** A well-known example comes from the evaluation of criminal recidivism prediction tools such as COMPAS. An investigative article by Angwin et al. (2016) showed that although the model the overall accuracy across racial groups was comparable, it displayed significant differences in false positive and false negative rates. Specifically, black defendants were more likely to be incorrectly labeled high risk, while white defendants were more likely to be incorrectly labeled low risk. Following this result, researchers examined these results in more detail, showing that notions of fairness such as calibration and equal error rates can be in contradiction with one another, and this effect is more pronounced when base rates are different within different sub groups (Chouldechova, 2017; Corbett-Davies et al., 2023). This case demonstrates how being aware of fairness during validation can uncover risks that would otherwise be overlooked. This example shows that careful metric selection is part of responsible model development rather than an optional after-thought.

**Adversarial robustness in image classification models:** A second example comes from the discovery of adversarial attacks in computer vision systems. Szegedy (2013) first demonstrated that adding very small, human-imperceptible changes to an input image can cause a neural network to make incorrect predictions with high confidence. Later, work showed that such vulnerabilities were widespread across many architectures and datasets (Goodfellow et al., 2014). These findings changed how models are evaluated in practice; instead of assessing performance only on clean test sets, researchers and practitioners now increasingly test

how models respond to perturbed or adversarial inputs. This shift has led to the development of models and training procedures that are more stable and reliable in realistic settings. Robustness testing during validation improves deployed systems without sacrificing good performance.

## 2.6. Model Deployment

This stage marks the point where the model crosses from development into real-world use. During deployment, the trained model interacts with live data, users, and operational systems. Decisions made at deployment have direct real-world consequences, so this phase requires careful validation, final checks, and clear accountability before the model goes live.

Main dimensions at this stage are **robustness and reliability** and **human oversight and accountability**. Also, privacy and security considerations in deployment include applying access controls and encryption before live usage, validating compliance with applicable privacy regulations and policies based on the deployment context and region. Moreover, ensuring transparency involves communicating the details, limitations, what the model is good at, and usage guidelines clearly, using language tailored to the intended end users.

Implementing fail-safes and fallback procedures should be incorporated as robust and reliable mechanisms to handle unexpected or anomalous inputs. Some mechanisms include default responses, alerting systems for human intervention, or routing high-risk cases to a human-in-the-loop, input validation checks, and rate-limiting mechanisms to prevent system overload.

Before deployment, clear roles for decision-making and intervention at launch should be assigned. Specific techniques to support this include defining meaningful human-in-the-loop gates for high-risk decisions, establishing escalation protocols for critical outcomes, and maintaining governance logs in terms of case inventories that track interventions, actions taken, and decision rationales. This stage should also ensure that processes are in place to address issues promptly, and responsibilities for governance, escalation, and remediation are clearly defined and confirmed, supporting accountability and responsible model operation from the moment the model goes live.

The following two examples show development and deployment performance gaps (first) and human oversight, transparency, and fairness failures (second).

**Epic sepsis prediction model:** A notable example of deployment challenges in real-world ML is the Epic sepsis prediction model, which was evaluated at Michigan Medicine across tens of thousands of hospitalizations. Despite strong performance metrics reported during development, independent evaluations after deployment revealed that the model

detected only about 33% of actual sepsis cases and produced a high false-alarm rate, which contributed to fatigue of clinician alertness and mistrust in the system (Habib et al., 2021). Critically, the deployment exposed a gap between research-level performance (e.g., AUROC on development data) and real-world clinical utility: the model often alerted clinicians too late to meaningfully change outcomes, and frequent false positives reduced trust and effective use in practice. This case demonstrates that performance in development datasets does not guarantee real-world effectiveness; responsible deployment requires rigorous external validation, transparent reporting of limitations, to ensure that intended benefits materialize in the target environment by aligning them with operational realities rather than isolated benchmark metrics.

**The Robodebt system in Australia:** Between 2016 and 2019, the Australian government deployed an AI-driven debt recovery system known as "Robodebt", which automatically issued debt notices to citizens based on algorithmic calculations of income discrepancies. By relying on automated income averaging without sufficient human oversight, the system produced numerous inaccurate debt notices, disproportionately affecting vulnerable welfare recipients (Michael, 2024). The Robodebt scandal underscores the critical importance of responsible choices in automated decision-making systems, particularly for government applications. The lack of human intervention in high-stakes decisions, combined with limited transparency and accountability, eroded public trust and caused widespread social and financial harm. Incorporating measures, such as fairness assessments, explainability standards, and structured human oversight, could have mitigated these ethical, legal, and downstream cost impacts.

### 2.7. Model Monitoring and Maintenance

The model monitoring and maintenance stage begins after deployment. Continuous monitoring helps ensure that the model continues to perform well over time and behaves responsibly when interacting with real people and real data under real conditions. This stage is especially important in high-stakes domains such as healthcare, finance, education, and employment.

A deployed model can fail for many reasons, with two common causes being data drift and concept drift (Lewis et al., 2022). Data drift occurs when the input data distribution changes, while concept drift occurs when the relationship between inputs and outcomes shifts. For example, consumer preferences may evolve, or the meaning of a "high-risk" pattern may change as social conditions shift. If these changes are not detected and addressed, model performance can slowly erode, harming accuracy, fairness, and reliability. Ongoing monitoring helps identify these issues early so that retraining, recalibration, or incident response can occur before performance degrades further.

Monitoring should also extend beyond just accuracy. Fairness metrics, subgroup performance, user experience, robustness, and resource use all provide important signals about system health. Which metrics matter most depends on the real-world goals of the system. For instance, in spam detection, users may prioritize a very low false-positive rate because missing important email is costly (Henke et al., 2015), while in loan approval, fairness and consistency across demographic groups may be central concerns. Responsible monitoring therefore relies on a set of thoughtfully chosen indicators rather than a single metric.

The two responsible AI principles that are especially important during the monitoring and maintenance stage are **accountability and auditability**, and **ongoing vigilance**. A responsible system should be open to review. This includes internal audits and, where feasible, external audits. In practice, this means keeping records of model updates, documenting changes in performance, and explaining how monitoring metrics are chosen and interpreted. When systems can be audited, organizations are more likely to detect emerging risks early and address them before real-world harm occurs. Fairness is not a one-time check. A model that is fair at launch may drift over time. Monitoring should therefore include subgroup performance and fairness-relevant indicators, rather than only global averages. This helps ensure that errors do not accumulate disproportionately for certain groups, and supports long-term equitable outcomes.

The next example shows how responsible monitoring and maintenance improve trust and model quality in real life.

**Detecting drift in healthcare AI systems:** Several deployed healthcare AI tools have shown performance degradation when used in new hospitals or changing patient populations. Finlayson et al. (2021) discusses how real-world deployment can expose models to shifts in clinical practice, coding standards, or patient demographics that were not reflected in the training data. In some cases, these shifts led to reduced diagnostic performance. Continuous monitoring helped identify these issues, prompting model retraining or recalibration. This example shows why responsible monitoring is essential in clinical settings, where overlooked drift may affect patient care.

## 3. Alternative Views

The alternative view to our argument is that there is an unavoidable tradeoff between a model's performance and responsible practices; in short, many practitioners believe that making a model more fair, secure, or private will make it less accurate. The origin of this belief is likely the result of early work that showed mathematical tradeoffs between

*Table 1.* Practitioner checklist across the ML lifecycle.

| Problem Formulation | Data Gathering & Pre-processing | Model Training |
|---|---|---|
| ☐ Conduct stakeholder & risk assessment
☐ Identify protected attributes and fairness criteria
☐ Document objectives, constraints, and key trade-offs
☐ Verify legitimacy and competency of the task | ☐ Audit dataset for representation gaps across subgroups
☐ Enforce consent and anonymization protocols
☐ Apply stratified sampling or targeted data enrichment
☐ Vet third-party data sources for bias and provenance
☐ Document all filtering and pre-processing decisions | ☐ Select loss functions and metrics aligned with fairness goals
☐ Apply in-process bias mitigation techniques
☐ Use interpretability tools (e.g., SHAP, LIME) during development
☐ Document hyperparameters, subgroup error rates, and constraints
☐ Log and version datasets, code, and model checkpoints |

| Validation & Evaluation | Deployment & Monitoring | |
|---|---|---|
| ☐ Evaluate accuracy, calibration, and subgroup fairness in parallel
☐ Stress-test on adversarial and out-of-distribution inputs
☐ Select metrics appropriate to domain consequences (e.g., recall in healthcare)
☐ Report known limitations and operational thresholds transparently
☐ Iterate with fixes when subgroup disparities are detected | ☐ Define human-in-the-loop gates for high-risk decisions
☐ Establish escalation protocols and governance logs
☐ Apply access controls and validate regulatory compliance
☐ Monitor continuously for data drift and concept drift
☐ Track subgroup fairness metrics post-deployment
☐ Schedule model retraining or recalibration as needed | |

different fairness metrics (Chouldechova, 2017; Kleinberg et al., 2016). In short, research showed that it was impossible to simultaneously equalize different performance metrics, such as false positive rate and accuracy, across different demographic groups. This came to be understood in the broader sense as a notion that imposing any kind of fairness constraint would also be giving up some kind of fairness. Recent works, however, have begun to challenge that perception (Kokhlikyan et al., 2022; Li et al., 2023; Shi et al., 2025). Specifically, Bell et al. (Bell et al., 2023) give a useful summary of work along this line and also show that the pure mathematical tradeoff that was originally proven does not manifest so strictly in practical settings.

Our position is, therefore, not that trade-offs never exist. Rather, we argue that responsible AI practices and model performance are not universally in competition. Navigating across the ML lifecycle, we show that integrating responsible AI frequently improves real-world system quality, particularly when performance is evaluated beyond narrow benchmark accuracy and instead includes realistic operating dimensions such as robustness, reliability, fairness, calibration, and deployability.

## 4. Call to Action

We argue for a shift in how responsible AI is framed and operationalized. Leadership plays a central role in shaping this shift. When responsible practices are implicitly positioned as slowing development or reducing accuracy, teams are discouraged from pursuing them. Leaders must instead make clear that responsibility and performance are aligned goals, and reflect this alignment in incentives, timelines, and success metrics.

Commitments to fairness and privacy must move beyond declarative statements. Responsible AI should be treated as a concrete performance objective and integrated into product conceptualization, data choices, model evaluation, and deployment decisions. Responsibility claims, especially on public forums like websites and white papers, should be supported by documented practices and measurable outcomes, not post hoc assurances.

To make this actionable, we offer a concise practitioner workflow and checklist organized by lifecycle stage in Table 1. Across all stages, three organizational practices can accelerate adoption. First, embed responsible AI criteria into existing definition-of-done checklists in engineering workflows, rather than treating them as parallel processes. Second, assign explicit ownership, someone accountable for fairness and robustness at each stage milestone, so these concerns are not deferred. Third, document trade-offs explicitly when they do arise: recording why a trade-off was accepted, and under what conditions it should be revisited, is itself a responsible practice that supports auditability and future improvement.

Finally, organizations should normalize cross-disciplinary collaboration throughout the development lifecycle. Input from domain experts, social scientists, and ethicists should inform problem formulation and risk assessment, particularly for systems with high potential for misuse. External consultation and independent auditing can further strengthen accountability. Treating responsible AI as a core component of system quality, rather than a constraint, is necessary for building models that are both effective and robust.

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
