# OpenReview forum: "Position: Responsible Practices and Model Performance are *Not* Competing Goals"
_ICML.cc/2026/Position_Paper_Track — ICML 2026 Position Paper Track regular_

### Official Review · Reviewer_u5W1 · 2026-03-09

**Significance:** 2
**Argument Clarity:** 2
**Rating:** 2
**Confidence:** 5

**Questions:**

Please refer to the details above.

**Alternative Views Section:**

Yes

**Compliance With Llm Reviewing Policy A Conservative:**

Affirmed.

**Discussion Potential:**

1

**Final Justification:**

Unfortunately, my concerns were not resolved, which were fundamental issues and the responses were what were already stated in the original submission.

Because the position of this paper has been already prevailing in the community, the position/perspective is not novel. I believe a position paper should argue for something important that is overlooked or underpracticed in the community. Hence, I am against accepting this position paper, and I maintain my score.

**Paper Summary:**

This paper argues that responsibility principles and model performance should not be competing goals. It addresses that the responsibility principles are critical objectives. It provides real-life instances for the argument.

**Position:**

Yes

**Position In Title:**

Yes

**Related Work:**

2

**Strengths And Weaknesses:**

The position that this paper argues is that the community has already taken it for granted because it is indeed true. As the authors mentioned, there exist papers that show that responsibility goals and performance are at odds, which tells us that those authors wish they would not be at odds, but they found it otherwise, and they alarmed the community through their research work. On the other hand, also as the authors mentioned, there are papers that show that there is a possibility to achieve responsibility goals not at the cost of model performance. Hence, this paper’s position is not novel, and I do not think we need this additional paper as a position paper.  Having said that, I understand the authors may have such a position, but then it should have been a technical research paper which develops methodologies/algorithms that can simultaneously achieve both responsibility goals and model performance, not as a position paper. Also, the authors mentioned earlier in the paper, they will show that responsibility goals and model performance are not at odds, but I do not believe the authors “showed/proved” it in this paper. (Having said that, again, even if they did it, it should have been a technical research paper.) For instance, through the Alternative view, the authors mentioned that the origin of the belief that the responsibility goals and model performance trade off with each other stems from the early work that showed it was impossible to simultaneously equalize different performance metrics, such as false positive rate and accuracy, across different demographic groups. In my opinion, if the authors had proposed a “fair” fairness metric to resolve the root cause, it would have been an important research paper as the authors already aware of the fundamental cause.

**Support:**

1

---

> ### Author Rebuttal · Authors · 2026-03-30
>
> We thank the reviewer u5W1 for this thoughtful feedback and would like to clarify the intended scope of our contribution. Our goal is not to introduce a new algorithm or propose a new metric. There is already a substantial body of technical work that studies trade-offs between different objectives, and we view our work as complementary to this line of research rather than a replacement for it.
>
> As the reviewer notes, prior work has analyzed trade-offs between specific metrics. In contrast, our unique contribution is to provide a unifying perspective across the ML lifecycle, bringing together evidence from real-world systems to show that, at different stages, responsible practices do not necessarily reduce overall system performance.
>
> We also clarify that we do not aim to prove that trade-offs never exist. Rather, our goal is to highlight that such trade-offs are often context-dependent and, in many practical settings, may be overstated through engineering design options or through advanced ML techniques like multi-objective optimization or through data augmentation.

---

> > ### Author Rebuttal · Reviewer_u5W1 · 2026-04-01
> >
> > Thank you for your responses. However, unfortunately, my concerns were not resolved, which were fundamental issues and the responses were what were already stated in the original submission.
> >
> > I have not asked the authors to introduce a new algorithm or propose a new metric. I am not sure whether the authors have thoroughly and correctly read my review comments. I only mentioned, "it should have been a technical research paper." in that regard.
> >
> > Again, because the position of this paper has been already prevailing in the community, the position/perspective is not novel. I believe a position paper should argue for something important that is overlooked or underpracticed in the community. Hence, I am against accepting this position paper, and I maintain my score.

---

### Official Review · Reviewer_e1Kd · 2026-03-13

**Significance:** 4
**Argument Clarity:** 3
**Rating:** 4
**Confidence:** 4

**Questions:**

- If responsibility can be placed in evaluation, then it can be part of model performance, rather than being a separate notion.

- Could you provide more concrete evidence showing how the mathematical tradeoffs do not show up in practical settings?

- Can you provide the case studies that are based on large language models?

**Alternative Views Section:**

Yes

**Compliance With Llm Reviewing Policy A Conservative:**

Affirmed.

**Discussion Potential:**

4

**Paper Summary:**

The authors argue that integrating responsible AI practices throughout the machine learning lifecycle improves, rather than hinders, overall model performance. The authors demonstrate their point through real-world case studies across all stages of ML development, which help improving downstream fauilures and making the AI models more robust and trustworthy.

**Position:**

Yes

**Position In Title:**

Yes

**Related Work:**

2

**Strengths And Weaknesses:**

Strengths:

- The authors breakdown the ML lifecycle and show the importance of responsible practices In each step to prevent error and real-world harms.

- The authors back up their claims with real-world case studies, showcasing the importance of the responsible AI.

Weaknesses:

- The authors mention recent research show that responsibility principles compete with model performance. However, the authors have not backed that claim with any recent papers.

- While the paper focuses on responsibility and model performance, those two notions are not clearly defined.

**Support:**

4

---

> ### Author Rebuttal · Authors · 2026-03-30
>
> We thank the Reviewer e1Kd for reviewing our work and for identifying key considerations to strengthen the position. Here are our more detailed responses and clarifications:
> 1. Definitions of Performance vs Responsibility -
> We will introduce clear operational definitions early in the paper or add a glossary in the appendix section in the camera-ready version. Here is a list of high-level definitions, which we plan to extend to the glossary:
>
>           Performance: Measures how effectively a model completes specific tasks. Common metrics include:
>                     Accuracy – the proportion of correct predictions.
>                     F1 Score – balances precision and recall for classification tasks.
>                     BLEU – evaluates the quality of generated text against reference outputs.
>
>          Responsibility: Captures the trustworthy and safe deployment of models, covering aspects such as:
>
>                    Fairness – minimizing bias against different groups.
>                    Safety – preventing harmful outputs.
>                    Robustness – maintaining performance under distribution shifts or adversarial inputs.
>                    Transparency – clarity of model behavior and decision-making.
>                    Privacy – protecting sensitive data.
>                    Alignment – ensuring outputs match human intentions or ethical norms.
>
>  Intersection of Performance and Responsibility: Some areas, like robustness and calibration, impact both performance and responsible deployment. Improving these aspects not only enhances trustworthiness but can also boost task-level effectiveness.
>
>
> 2. Engagement with Trade-off Literature
>
> We will add both theoretical and empirical references:
>
>           a. Theoretical trade-off (robustness–accuracy): Zhang et al., 2019 (ICML).
>
>           b. Empirical evidence:
>
>                    i. Public-sector ML shows fairness–accuracy trade-offs are negligible: Rodolfa et al., 2021.
>
>                    ii. Clinical ML bias mitigation improves fairness while preserving performance: Yang et al., 2023.
>
>       [1] Rodolfa, K.T., Lamba, H., \& Ghani, R. (2021). Empirical observation of negligible fairness–accuracy trade-offs in machine learning for public policy. \textit{Nat Mach Intell}, 3, 896–904. \url{https://doi.org/10.1038/s42256-021-00410-x}
>
>     [2] Yang, J., Soltan, A.A.S., Eyre, D.W., et al. (2023). Algorithmic fairness and bias mitigation for clinical ML with deep reinforcement learning. \textit{Nat Mach Intell}, 5, 884–894. \url{https://doi.org/10.1038/s42256-023-00711-2}
>
> We will also explain why theoretical trade-offs often do not manifest due to overparameterization, data distribution shifts, and multi-objective optimization regimes.
>
> 3. Large Language Model Case Studies -
>
> We will add 1--2 focused LLM case studies:
>
>      a. Alignment interventions: Instruction tuning or safety fine-tuning improves helpfulness/safety metrics while maintaining or improving benchmark performance: Ouyang et al., 2022.
>
>      b. Bias mitigation techniques: In-training vs post-training approaches. In-training methods are more effective at reducing social biases (gender, race, religion) while maintaining strong performance: Arzaghi et al., 2025.
>
>      [3] Ouyang, L., Wu, J., Jiang, X., et al. (2022). Training language models to follow instructions with human feedback. \textit{NeurIPS}, 35, 27730–27744. \url{https://proceedings.neurips.cc/paper/2022/hash/1f89885dcb68d3eac7a2ff197f66c4a6-Abstract.html}
>
>      [4] Arzaghi, M., Farashah, A.D., Carichon, F., \& Farnadi, G. (2025). Intrinsic meets extrinsic fairness: Assessing the downstream impact of bias mitigation in large language models. \textit{arXiv:2509.16462}. \url{https://arxiv.org/abs/2509.16462}

---

> > ### Author Rebuttal · Reviewer_e1Kd · 2026-04-04
> >
> > I would like to thank the authors for their response and for addressing some of my points.
> >
> > I think my first question was not addressed properly by the authors and I do not think the paper considers that case at all.
> > The authors continue to treat performance and responsibility as two separate concepts. In current times, responsibility can be a performance metric as well, such as safety, fairness, or robustness. We would define a model to be performing well if it satisfies the earlier notions (apart from the typical benchmark performance). An irresponsible model is actually an underperforming model, in that case.
> >
> > If the authors took a more modern position, the paper would be more impactful.

---

### Official Review · Reviewer_2xP3 · 2026-03-17

**Significance:** 2
**Argument Clarity:** 3
**Rating:** 5
**Confidence:** 4

**Questions:**

See above.

**Alternative Views Section:**

Yes

**Compliance With Llm Reviewing Policy A Conservative:**

Affirmed.

**Discussion Potential:**

2

**Final Justification:**

Thank you for the clarification, which basically addressed my concerns, I will keep my score.

**Paper Summary:**

This work argued that responsibility and performance of ML system are not competent. Starting from a lifecycle of ML system, this work discussed the responsible AI practices at each stage, with examples detailing the benefit of integration of responsibility into the ML system design. Also the work briefly discussed the alternative view that the tradeoff between responsibility and performance is unavoidable. Finally the work calls to action for leadership to make responsible AI into the workflow of the full cycle of ML system.

**Position:**

Yes

**Position In Title:**

Yes

**Related Work:**

3

**Strengths And Weaknesses:**

Strength:
1. The topic of this work is closely related ML community, and the challenge on the common tradeoff viewpoint can inspire certain discussion.
2. There are plenty of real-world examples with details throughout the work, which can help the reasoning of the work.
3. The writing of the work is easy to follow.

Weakness:
1. Some details should be clarified to support the "not in competition" main claim of the work. For example, the adversarial robustness example in Section 2.5, generally such robust optimization models will lead to a degrade on the accuracy, while increasing the model robustness (i.e., tradeoff), such "without sacrificing good performance" statement should be supported with detailed references if any.
2. It would be more convincing to have more examples with "before" and "after" treatments. I prefer the X-ray model equity example in that you provided detailed performance data showcasing that the robust metrics improved, while without sacrificing the model performances in general. But for other examples mentioned in this work, they generally lacks such data and results, or just come with logical reasoning.

**Support:**

3

---

> ### Author Rebuttal · Authors · 2026-03-30
>
> We thank Reviewer 2xP3 for their careful review and the thoughtful suggestions provided. Here are our detailed responses:
> 1. Clarifying the Central Claim: Responsibility--Performance Relationship
>
> We agree that our original phrasing may have appeared overly strong. Our intended claim is not that trade-offs never exist, but that responsibility and performance are not universally in competition. In the revision, we will:
>
>           i. Soften the claim to acknowledge that trade-offs may arise in some settings (e.g., adversarial robustness or strict fairness constraints).
>
>           ii. Clarify conditions where alignment is more likely than a trade-off, such as:
>
>                            a. Responsible practices reducing spurious correlations or distributional mismatch.
>
>                            b. Improved data quality or evaluation design, enhancing both reliability and accuracy.
>
>                            c. Considering system-level performance (safety, generalization, calibration) beyond single-metric accuracy.
>
>                            d. Multi-objective optimization in modern ML.
>
> A dedicated paragraph in the introduction will formalize this nuanced position.
>
> 2. Adversarial Robustness Example and Supporting Citations
> We will strengthen the section with:
>
>          i. Trade-off citation: Zhang et al., 2019. \textit{Theoretically Principled Trade-off between Robustness and Accuracy}, ICML 36, 7472–7482. \url{https://proceedings.mlr.press/v97/zhang19p.html}
>
>          ii. Empirical evidence where robustness interventions improve performance: Sanaat et al., 2022. \textit{Robust-Deep: Increasing brain imaging datasets to improve performance and robustness}, Journal of Digital Imaging, 35(3), 469–481.
>
> We will clarify that robustness trade-offs are context-dependent and may be mitigated via data augmentation, architecture scaling, and distribution-aware evaluation.
>
> 3. Strengthening Empirical Evidence (“Before vs. After” Examples)}
> We will add 2-3 additional quantitative case studies, similar to the X-ray equity example, providing “before vs. after” comparisons to support the performance-aligned claim.

---

> > ### Author Rebuttal · Reviewer_2xP3 · 2026-04-07
> >
> > Thank you for the response, which helps to clarify the strength of the some claims, also added more supportive citations, I believe the work should benefit a lot from the promised revision in the future version. I will keep my score.

---

### Official Review · Reviewer_FQTr · 2026-03-18

**Significance:** 3
**Argument Clarity:** 3
**Rating:** 5
**Confidence:** 3

**Questions:**

N/A

**Alternative Views Section:**

Yes

**Compliance With Llm Reviewing Policy A Conservative:**

Affirmed.

**Discussion Potential:**

3

**Final Justification:**

The authors addressed the issues I raised.
I find their plan to sharpen their Call for Action quite helpful: proposing a concrete workflow across the ML lifecycle and checklists and guidance on likely trade-off are a very valuable addition.

**Paper Summary:**

The authors provide various arguments and examples showing the consequences of ignoring responsible AI practices and how mitigating them does not necessarily affect the model's accuracy.

**Position:**

Yes

**Position In Title:**

Yes

**Related Work:**

3

**Strengths And Weaknesses:**

Strength:
- The paper is well written and easy to follow.
- Addressing multiple aspects of responsible AI, including fairness, privacy preservation, and security.
- Tangible real-world examples from various domains showing the impact of ignoring the above-mentioned responsible AI dimensions.
- Examples showing how those aspects can be addressed with minor impact on model's accuracy.

Weaknesses:
- Lack of theoretical framework showing how different objectives can be met together to counteract the narrow study of (Chouldechova, 2017; Kleinberg et al., 2017). Multiple studies have shown that e.g. improving fairness does improve model accuracy and generalization, by forcing the model to work harder instead of naively relying on protected attributes as lazy shortcuts to draw decision boundaries (see Kokhlikyan et al. for example).
- Call for action is vague and not concrete.

Kokhlikyan, Narine, et al. "Bias mitigation framework for intersectional subgroups in neural networks." NeurIPS 2022 Workshops (arXiv preprint arXiv:2212.13014).

Wording issues:
- netflix => Netflix
- first-class pre-processing constraint => primary ("first class" is not a scientific term).

**Support:**

3

---

> ### Author Rebuttal · Authors · 2026-03-30
>
> We thank Reviewer FQTr for reviewing our paper and providing valuable suggestions. Our detailed responses and clarifications are as follows:
>
> 1. On literature about fairness improving model accuracy and generalization: We thank the reviewer for pointing out this important line of study and will incorporate it more explicitly in the revision.
>
> Multiple studies have shown that improving fairness can also improve model accuracy and generalization, for example by encouraging models to rely less on protected attributes as shortcuts and instead learn more robust decision boundaries (e.g., Kokhlikyan et al. [1]). More broadly, recent work shows that fairness interventions can improve both predictive accuracy and generalization, for instance by improving feature representations or reducing reliance on spurious correlations [2–3], suggesting that fairness can act as a form of regularization rather than a constraint.
>
>       [1] Kokhlikyan, Narine, et al. "Bias mitigation framework for intersectional subgroups in neural networks." NeurIPS 2022 Workshops (arXiv preprint arXiv:2212.13014).
>
>       [2] Li, Xuran, et al. "Accurate fairness: Improving individual fairness without trading accuracy." In Proceedings of the AAAI Conference on Artificial Intelligence (Vol. 37, No. 12, pp. 14312-14320). 2023
>
>       [3]  Shi, Yingdong, et al. "Dissecting and mitigating diffusion bias via mechanistic interpretability." Proceedings of the Computer Vision and Pattern Recognition Conference. 2025.
>
> 2. On strengthening the call to Action:  We appreciate this feedback and will strengthen the call to action in the revision. In particular, we will include a concrete workflow across the ML lifecycle, along with practitioner-oriented checklists and clearer guidance on when trade-offs are likely to arise versus when they can be mitigated in practice.

---

> > ### Author Rebuttal · Reviewer_FQTr · 2026-04-02
> >
> > I appreciate the author's response and their plan to sharpen their Call for Action.
> > I find the workflow across the ML lifecycle and the proposed checklists and guidance on likely trade-off a very valuable addition.
> >
> > I am raising my score accordingly.

---

### Decision · Program_Chairs · 2026-04-30

**Decision:**

Accept (regular)

**Comment:**

The paper argues for an important position that is substantiated through real-world examples across different stages of the lifecycle of an ML model. The paper is well-written and easy to follow and is likely to spark important discussions about the role of responsible practices when building ML models.

Reviewers brought up concerns including missing citations (e.g work showing that improving fairness improves accuracy and generalization), potential confusion around the scope of the claim, given that in some cases performance and responsibility are indeed competing, the need for a clearer definition of the terms “performance” and “responsibility”, as well as novelty concerns.

During the rebuttal, the authors clarified that their position is not that trade-offs never exist, but rather that they don’t always exist, discussing appropriate examples. They committed to clarifying this important issue and adjusting claims where needed, and they also discussed the conditions under which the alignment between performance and responsibility is more likely to occur, which they laid out in their rebuttal. The authors also discuss related work that will be discussed in the manuscript to address reviewers’ suggestions.

All reviewers signaled that their concerns were addressed during the rebuttal, except for Reviewer u5W1 who raised concerns about the novelty of the position, given that prior work has already identified this phenomenon in different settings. However, the authors clarified that the nature of their contribution is to offer a unifying perspective across the lifecycle of an ML model, synthesizing evidence from real-world systems and across different stages.

I believe that the existence of prior technical work that has shown synergies between performance and responsibility in certain isolated scenarios does not lessen the degree of contribution of this position paper, which actually builds upon and unifies those observations, discusses how these issues manifest in different phases of a model’s lifecycle, and makes calls for action for the community that are likely to spark important discussions on this timely issue.